# In Planta, In Vitro and In Silico Studies of Chiral *N*^6^-Benzyladenine Derivatives: Discovery of Receptor-Specific *S*-Enantiomers with Cytokinin or Anticytokinin Activities

**DOI:** 10.3390/ijms231911334

**Published:** 2022-09-26

**Authors:** Ekaterina M. Savelieva, Anastasia A. Zenchenko, Mikhail S. Drenichev, Anna A. Kozlova, Nikolay N. Kurochkin, Dmitry V. Arkhipov, Alexander O. Chizhov, Vladimir E. Oslovsky, Georgy A. Romanov

**Affiliations:** 1Timiryazev Institute of Plant Physiology, Russian Academy of Sciences, Botanicheskaya str. 35, 127276 Moscow, Russia; 2Engelhardt Institute of Molecular Biology, Russian Academy of Sciences, Vavilov str. 32, 119991 Moscow, Russia; 3Zelinsky Institute of Organic Chemistry, Russian Academy of Science, Leninsky pr. 47, 119991 Moscow, Russia

**Keywords:** cytokinin, anticytokinin, chirality, *R*-, *S*-enantiomers, AHK receptor, receptor specificity, 6-benzyladenine derivatives, nucleobase, ribonucleoside

## Abstract

Cytokinins, classical phytohormones, affect all stages of plant ontogenesis, but their application in agriculture is limited because of the lack of appropriate ligands, including those specific for individual cytokinin receptors. In this work, a series of chiral *N*^6^-benzyladenine derivatives were studied as potential cytokinins or anticytokinins. All compounds contained a methyl group at the α-carbon atom of the benzyl moiety, making them *R*- or *S*-enantiomers. Four pairs of chiral nucleobases and corresponding ribonucleosides containing various substituents at the *C*2 position of adenine heterocycle were synthesized. A nucleophilic substitution reaction by secondary optically active amines was used. A strong influence of the chirality of studied compounds on their interaction with individual cytokinin receptors of *Arabidopsis thaliana* was uncovered in in vivo and in vitro assays. The AHK2 and CRE1/AHK4 receptors were shown to have low affinity for the studied *S*-nucleobases while the AHK3 receptor exhibited significant affinity for most of them. Thereby, three synthetic AHK3-specific cytokinins were discovered: *N*^6^-((*S*)-α-methylbenzyl)adenine (*S*-MBA), 2-fluoro,*N*^6^-((*S*)-α-methylbenzyl)adenine (*S*-FMBA) and 2-chloro,*N*^6^-((*S*)-α-methylbenzyl)adenine (*S*-CMBA). Interaction patterns between individual receptors and specific enantiomers were rationalized by structure analysis and molecular docking. Two other *S*-enantiomers (*N*^6^-((*S*)-α-methylbenzyl)adenosine, 2-amino,*N*^6^-((*S*)-α-methylbenzyl)adenosine) were found to exhibit receptor-specific and chirality-dependent anticytokinin properties.

## 1. Introduction

The cytokinin signaling system is one of the most important regulatory systems in all land plants, including evolutionarily ancient species [1,2,3,4,5,6]. It functions at every stage of plant ontogenesis, primarily regulating cell proliferation and differentiation [7,8,9,10]. However, this complex and versatile cytokinin action often precludes the wide practical application of this phytohormone in crop production. For example, the positive effect of cytokinins on shoot growth (stimulation of lateral meristem and cambium growth as well as chloroplast differentiation [11,12]) is often compromised by a negative effect on root growth [12,13].

One of the ways to overcome this problem is the use of ligands specific to shoot-located individual cytokinin receptors. Cytokinin receptors of any plant species usually constitute a family of several structurally similar transmembrane proteins with histidine kinase activity [14,15,16,17]. Despite their structural similarity, receptors of the same family often have differences in ligand specificity, which seems to be related to their participation in long-distance root-to-shoot signaling [8,15,16]. In the model plant Arabidopsis, particularly, three cytokinin receptor isoforms (AHK2, AHK3 and CRE1/AHK4) are distributed unevenly in plant organs: AHK3 dominates in the shoot, while CRE1/AHK4 dominates in the root [8,18]. Thus, the use of receptor-specific ligands may lead to local cytokinin effects on selected organs and tissues.

The empirical search for new cytokinins with desired properties has typically been not very effective. Nevertheless, it has helped to identify some interesting compounds with either cytokinin [19,20,21,22,23,24,25,26,27] or anticytokinin [28,29,30,31] activity. The assay systems we developed to screen substances for cytokinin activity [32,33,34,35] have contributed to the discovery of new promising ligands that interact with cytokinin receptors. Such screenings of large series of compounds has allowed direct detection of receptor-specific ligands possessing cytokinin or anticytokinin properties. These findings expand and deepen our knowledge of the ligand–AHK receptor interaction and strengthen an experimental basis that is also useful in the applications of molecular modeling.

Previously, we found a compound 2-chloro,*N*^6^-(α-methylbenzyl)adenine (2-chloro-6-(1-phenylethyl)adenine), that selectively interacted with AHK3, one of the three cytokinin receptors of Arabidopsis [26]. Although certain structural features of synthetic cytokinin molecules affecting their interaction with individual receptors have been noted, the basic nature of such specificity has remained unclear. Additional difficulties in understanding the patterns of cytokinin–receptor interaction arise from the fact that some ligands can be a mixture of *R*- or *S*-enantiomers that differ in spatial structure. This was particularly the case of 2-chloro,*N*^6^-(α-methylbenzyl)adenine which represented a mixture of *R*- and *S*-enantiomers in an unknown proportion.

Enantiomers (optical isomers) are stereoisomers whose structures are mirrored to each other. Their structures are non-identical because of the different spatial arrangement of the substituents around the optically active (chiral) carbon atom. Therefore, ligands-enantiomers can bind with receptors in different ways. In nature, macromolecules are homochiral, since proteins and enzymes in the cells of all living organisms consist of L-amino acids (except achiral glycine), which correspond to *S*-enantiomers, while DNA and RNA contain monomeric carbohydrate units in the D-configuration, which correspond to *R*-enantiomers. Furthermore, the vast majority of natural monosaccharides belong to the D-configuration. At the same time, there are examples when L-isomers (L-Rha, L-Fuc, L-Ara) are the most common in nature; for example, L-Ara is more common in plants, while D-Ara is found in some species of microorganisms [36,37,38]. Chirality plays an extremely important role in most biochemical processes, because it affects the spatial arrangement of macromolecules and determines the convergence and interaction of active groups in enzyme–substrate complexes [37,39,40], which ensures the selectivity of biochemical processes occurring in the cell [41].

To the best of our knowledge, to date no systemic studies of the chirality effect on the ligand interaction with cytokinin receptors have been performed. In the present work, we investigated individual *R*- and *S*-enantiomers − *N*^6^-benzyladenine (BA) derivatives. We found that the chirality of the ligand molecule has a strong influence on the affinity of the potential cytokinin for the cognate receptor. In our study, the chirality affected the manifestation of both cytokinin and anticytokinin activity of the BA nucleobase/ribonucleoside derivatives. Some of newly uncovered *S*-enantiomers-nucleobases with cytokinin activity exhibited strong preference for AHK3 receptors. At the same time, two *S*-enantiomers-ribosides were shown to possess selective anticytokinin activity toward AHK2 and CRE1/AHK4, but not AHK3 receptors.

## 2. Results and Discussion

### 2.1. Chemical Synthesis

For the synthesis of chiral derivatives of the potent cytokinin *N*^6^-benzyladenine (BA), some known and modified methods were used. The compounds were obtained starting from 6-activated purine derivatives during nucleophilic substitution reaction by secondary optically active amines (total 16 compounds).

#### 2.1.1. Synthesis of *N*^6^-Alkyladenines

To modify natural cytokinin *N*^6^-benzyladenine (BA) molecule with the α-methylbenzyl chiral fragment, various 6-chlorosubstituted purines were introduced into amination reaction with *R*- and *S*-(α-methylbenzyl)amine in the presence of Hunig’s base DIPEA (Figure 1) as was previously shown for non-chiral purine derivatives [19,20]. Details of the synthesis are described in the Section 4.

#### 2.1.2. Synthesis of Ribonucleosides

*N*9-Ribosylated forms of cytokinins represent their metabolic precursors releasing active hormones by distinct biochemical pathways [1,42]. To study the biological activity of possible metabolites of chiral cytokinin analogues, a series of their ribosylated forms was synthesized as shown in Figure 2, Figure 3, Figure 4 and Figure 5. Details of the synthesis are described in the Section 4.

Finally, 16 enantiomers were synthesized and purified, representing derivatives of adenine and adenosine, in other words, 8 nucleobases and 8 ribonucleosides. Each basic series included compounds with the same modifications at *C*2 position of the purine heterocycle, where hydrogen (H) can be substituted either with fluorine (F), chlorine (Cl) atoms or amino group (NH_2_) (Table 1).

For all obtained compounds in all cases both *R*- and *S-*enantiomers could not be distinguished by NMR or UV-spectra as these types of spectra were almost identical (see Appendix A). This was due to the similarity of the structure and chemical properties of the α-methylbenzyl fragment in *R*- and *S*-configurations present in the molecule: as a result, the difference in chemical shifts or absorption maxima were insignificant. To distinguish between *R*- and *S*-enantiomers, the circular dichroism method (CD) was applied along with the NMR technique, exemplified by CD spectra of compounds **12a** and **12b** (Appendix A).

#### 2.1.3. Synthesis of Ado^BOM^ and Ade^BOM^

To study the anticytokinin activity of adenine derivatives bearing hydrocarbon fragments, some particular BA derivatives were obtained as reference compounds (Figure 6). The first of these was *N*^6^-(benzyloxymethyl)adenosine (Ado^BOM^, BOMA in [31], a BA riboside which has been shown to exhibit anticytokinin properties with the receptor CRE1/AHK4. The cognate free nucleobase *N*^6^-(benzyloxymethyl)adenine (Ade^BOM^) was also synthesized. Details of both syntheses are described in the Section 4.

### 2.2. Study of the Biological Activity of the Chiral Compounds

#### 2.2.1. Cytokinin Activity of Compounds in Arabidopsis Bioassays

For the entire series of novel BA derivatives under study (both free nucleobases and ribonucleosides, see Table 1), we evaluated their cytokinin activity at a concentration of 1 µM in a bioassay with double receptor mutants of Arabidopsis, each mutant retaining a single cytokinin receptor. All Arabidopsis plants used in the bioassay harbored the *GUS* reporter gene fused with the promoter of the cytokinin primary response gene (*ARR5*) [26,43]. Thus, the hormonal activity of the studied compounds was defined as GUS activity, representing the expression intensity of the *P_ARR5_:GUS* construction (Figure 1).

Cytokinin *N*^6^-benzyladenine (BA) at the same concentration (1 µM) and plain water were used as positive and negative controls, respectively. The GUS activity of BA was taken as 100%, the activity of other compounds was determined in % relative to BA activity. The background level of GUS activity (negative control values) was quantified and subtracted from all values obtained.

According to their cytokinin activity relative to that of BA measured in bioassays (Appendix A), all compounds can be classified into three conditional subgroups: with no or low (less than 15% of the BA level), medium (from 15 to 85% of the BA level), and high (above 85% of the BA level) cytokinin activity. All of the *R*-nucleobases (at a concentration of 1 µM) had moderate to strong cytokinin activity with all Arabidopsis cytokinin receptors (Figure 1). It should be noted that the activity of *R*-nucleobases toward receptors mainly did not change when chlorine atom was a substituent (except a marked decrease in AHK2 variant) or even increased (by an average of 40%) with fluorine atom at the *C*2 position (Figure 1A). We have shown earlier that the insertion of halogen at the adenine *C*2 position can increase the activity of various BA-derivatives in their interaction with the AHK3 receptor [26]. In this work, a similar statistically significant effect was observed for all Arabidopsis cytokinin receptors, especially for AHK3 and CRE1/AHK4 (Figure 1A).

Of particular interest are data shown in Figure 1B where the effects of *S*-nucleobases are demonstrated. Cytokinin receptor AHK3 was able to actively interact with three of the four *S*-bases (at 1 µM concentration) while AHK2 and CRE1/AHK4 receptors were almost unable to. Thus, we found three compounds of a similar structure specific to the AHK3 receptor—**5b**, **6b**, and **7b**. We have previously shown [26] that a mixture of 2-chloro,*N*^6^-(α-methylbenzyl)adenine (2-chloro-6-(1-phenylethyl)adenine) enantiomers (**7a** and **7b** in this study) at 1 µM concentration preferentially activated the AHK3 receptor over the other two. Apparently, the *S*-enantiomer prevailed in the mixture used in the previous study, otherwise it would be not possible to detect such receptor specificity. As for enantiomers **5b** and **6b**, these compounds are described here for the first time as selective activators of the AHK3 receptor. These *S-*enantiomers share a similar structure, differing only in the substituent at the *C*2 atom of the purine heterocycle: molecules **5b**, **6b** and **7b** have H, F or Cl substituents, respectively. These results demonstrate the principal possibility of creating receptor-specific hormones, at least for the Arabidopsis cytokinin receptor family. Compounds **5b**, **6b** and **7b** can be shortly named as *S-***MBA** (*N*^6^-(α-**m**ethyl**b**enzyl)**a**denine)), *S**-*****FMBA** (2-**f**luoro,*N*^6^-(α-**m**ethyl**b**enzyl)**a**denine) and *S***-CMBA** (2-**c**hloro,*N*^6^-(α-**m**ethyl**b**enzyl)**a**denine), respectively. The amino group as a substituent at *C*2 appears to be unfavorable for the receptor’s ability to bind such a ligand, and its presence causes the compounds to diminish activity or totally abolish it, as in the case of *S*-isomer interaction with the AHK3 receptor (Figure 1A,B and Appendix A).

Overall, Appendix A shows a clear parallelism between the AHK2 and CRE1/AHK4 receptors in the preference for the chiral compounds under study, whereas the receptor AHK3 is quite different in this respect. This is consistent with our previous data on the similarity of the ligand specificity between AHK2 and CRE1/AHK4 receptors [16,43].

We visualized the action of uncovered receptor-specific compounds on Arabidopsis seedlings by histochemical tissue staining in situ (Figure 2). For this purpose, compound **7b** (*S*-**CMBA**) was selected, which has been already supposed in [26] to be responsible for the receptor-specific activity.

Histochemical staining data confirmed that *S*-**CMBA** strongly affected AHK3-expressed mutants but generally had no or little effect on seedlings with single AHK2 or CRE1/AHK4 receptors. Plants with the latter receptors treated with *S*-**CMBA** looked like plants incubated in plain water. The only difference noticed in this case was stronger staining of the roots of seedlings expressing CRE1/AHK4 receptor, resembling the root staining in the positive control variant. It should be noted that the CRE1/AHK4 receptor is characterized as a root receptor [16], expressing predominantly in roots. The bulk of cytokinins are known also to be synthesized in roots. Therefore, the elevated background staining of roots in CRE1/AHK4 expressing mutants was a consequence of the enhanced endogenous cytokinin signaling. The apparent stronger root staining of *S*-**CMBA**-treated plants versus mock-treated ones may result either from the direct effect of the compound or from its indirect influence through endogenous cytokinin content, or both.

The cytokinin activity of the chiral ribonucleosides (ribosides) at 1 µM concentration was negligible and hardly dependent on the chirality of the molecule (Figure 1C,D and Appendix A). Among the *S*-ribonucleosides, none of the compounds showed any significant cytokinin activity, hence, addition of riboside residue at *N*9 rendered these BA derivatives fully inactive. Almost all *R*-ribonucleosides behaved similarly except *N*^6^-((*R*)-1-α-methylbenzyl)adenosine (**12a**) which exhibited rather low but detectable activity (23.9%) with only AHK3 receptor.

Thus, the activity of ribonucleosides in the biotests in almost all variants drastically dropped as compared to cognate free bases. This is consistent with the notion that only cytokinin bases have real biological activity, whereas their ribosides are mostly the inactive transport forms of these hormones and either cannot bind to cytokinin receptors or, even when binding, are unable to activate them [44]. Although ribonucleoside **12a** demonstrated some GUS activity, mainly with the AHK3 Arabidopsis receptor, and the activity of cytokinin ribosides in biotests has been previously described [45], this residual activity was presumably due to their enzymatic conversion to free bases in vivo (see also the discussion below).

The structure of the heterocyclic ribonucleoside base undoubtedly affects the efficiency of ribose cleavage [45]. In this work, we also observed that the presence of some substituents at *C*2 apparently made it difficult to detach the ribofuranose residue, which blocked the cytokinin activity of riboside compound. Given the difference in the activities of the riboside isomers **12a** and **12b**, we can assume that the chirality of the compound may affect its recognition by the corresponding enzyme or enzymes.

Thus, the chirality greatly affects the properties of cytokinin-derived enantiomers. *R*-isomers (bases) were quite active at 1 μM concentration but largely unspecific regarding structural peculiarities and different receptors, whereas *S*-isomers, being generally less active, were evidently more sensitive to small structural changes in their structure and, most important, exhibited high receptor specificity interacting almost exclusively with the AHK3 receptor (Figure 1 and Figure 2 and Appendix A).

Our study aimed, among other purposes, at searching for compounds with anticytokinin action, i.e., capable of inhibiting the receptor activation by cytokinin. In the infrequent studies describing modified adenine derivatives with anticytokinin properties [28,29,31], the specific action of such compounds was observed at rather high concentrations (10–50 µM). For example, the suppression of CRE1/AHK4 receptor activation by anticytokinin Ado^BOM^ was tested at its 500-fold excess (50 µM Ado^BOM^ versus 0.1 µM BA) [31]. At the same time, in all the above anticytokinin studies, the virtual lack of cytokinin activity of anticytokinins even at high (up to 50 µM) concentrations was noted. Therefore, we also assayed the chiral compounds (except *R*-nucleobases) for the GUS activity at a concentration 50 µM (Table 2). In this case, the *R*-nucleobases seemed not suitable as potential anticytokinins because they displayed pronounced cytokinin activity even at a concentration as low as 1 µM.

To further test for anticytokinin activity, we selected compounds exhibiting at 50 µM weak or no activity (less than 15% of the 1 µM BA level). For the AHK3 receptor, no such compounds were found in our study. In contrast, the AHK2 and CRE1/AHK4 receptors gave similar patterns of activity, and in both cases the *S*-ribonucleosides **12b**, **13b**, **14b**, **16b** had little or no activity (Table 2).

#### 2.2.2. Anticytokinin Activity of Compounds in Arabidopsis Bioassay

The compounds selected in previous experiments were tested for their anticytokinin activity. For this purpose, these compounds, at a concentration of 50 µM, were mixed with 0.1 µM BA and the activity of the mixture was compared with the activity of 0.1 µM BA solution alone (100% control) in the *P_ARR5_:GUS* Arabidopsis biotest. When the activity of the mixture was lower than that of the BA control (i.e., reliably below 100%), the compound was classified as exhibiting anticytokinin properties. In total, eight compounds were tested for anticytokinin activity, and two of them (**12b** and **16b**) significantly inhibited (by ~30%) BA activity with individual AHK2 and/or CRE1/AHK4 receptors (Table 3). In addition, we tested *S*-ribonucleosides **12b** and **16b** for their anticytokinin activity with AHK3 receptor. Free *S-*nucleobases **5b** and **8b** (corresponding to ribosides **12b** and **16b**) and *R*-ribosides **12a** and **16a** (enantiomers of **12b** and **16b**) were also checked for the presence of anticytokinin activity with all three receptors (Table 3).

Compounds **12b** and **16b** demonstrated clear anticytokinin activity. At the same time, no other *S*-ribonucleosides containing fluorine or chlorine atom in the *C*2 position nor cognate *R*-enantiomers had anticytokinin effect.

Consequently, the anticytokinin effect seems to be not inherent to all ribosides of BA-derivatives and not to all *S*-ribonucleosides. Based on the data obtained, we can assume that the manifestation of anticytokinin properties depends not only on the chirality of the optically active C-atom in the *N*^6^ side chain but also on the *C*2 substituent in the purine structure.

There is very little information on the detection of anticytokinins [28,29,30,31]. Of the four described anticytokinins, three compounds are BA derivatives: two free bases and one ribonucleoside. The nucleobases PI-55 [28] and LGR-991 [29] differ from BA only by substituents at the phenyl radical. In contrast, the ribonucleoside Ado^BOM^ [31] has an altered structure of the linker between the adenine heterocycle and the phenyl radical as compared to BA. According to [31], Ado^BOM^ specifically suppressed, by 50–60%, the cytokinin activation of the receptor CRE1/AHK4. Thus, compounds **12b** and **16b** are very similar to Ado^BOM^ in their structure, effect in the biotest, and receptor specificity. Given the noted information paucity about the anticytokinin detection, the explanation for the action of two new similar compounds at once apparently should be sought in the structural features that unite them with Ado^BOM^. For this purpose, we included Ado^BOM^ in our further experiments. In addition, a free base (Ade^BOM^) corresponding to Ado^BOM^ was synthesized. The results of the experiments with Ado^BOM^ and Ade^BOM^ are given and discussed below.

#### 2.2.3. (Anti)Cytokinin Binding Assay

It is commonly assumed that the direct activation (inactivation) of a receptor by its agonist (antagonist) is based on a high-affinity ligand–receptor interaction. The results presented in Table 2 show that many of the tested ribonucleosides exhibited medium or even strong hormonal activity at high (50 µM) concentration. The question arose as to whether this response in the GUS activity biotest was really caused by direct binding of these ligands to the receptor or was due to other reasons, for example, the metabolic conversion of ribosides.

We assessed the specific binding of BA derivatives to each of the Arabidopsis cytokinin receptors by radioligand competition assay. Tritium-labeled isopentenyladenine (^3^H-iP) was used as a labeled ligand. The plant assay system based on plant microsomes that carried transiently expressed receptors [34,35] was used. Ribonucleosides and *S*-nucleobases were tested at concentrations of 1 and 50 µM, while *R-*nucleobases were tested only at 1 µM (Table 4). The latter strongly bound to the receptors already at a concentration 1 µM, therefore we considered it meaningless to also measure the binding of *R-*nucleobases at a 50-fold higher concentration. The strong binding of *R*-bases was largely non-selective toward different receptors (Table 4) and correlated well with the in planta transcriptional effects of these compounds (Figure 1, Appendix A).

A similar correlation may be extended to *S*-nucleobases (1 µM) as well. Here maximum binding and maximum effect were observed when testing *S*-enantiomers **6b** and **7b** with the AHK3 receptor, whereas receptor AHK2 and *S*-base **8b** showed no binding and no effects in any of the experimental probes. The level of binding of all *S*-nucleobases to each Arabidopsis cytokinin receptor was significantly increased along with an increase in their concentration from 1 to 50 µM. In several probes, this increase was quite large. The AHK3 retained its advantage over CRE1/AHK4 and AHK2 receptors in binding compounds **5b** and **8b**, but almost lost this advantage for the *S*-bases **6b** and **7b** (Table 4). Again, these data are consistent with corresponding GUS activity data displayed in Figure 1 and Appendix A. Hence, both *R*- and *S*-nucleobases demonstrated clear correlations between binding affinity to receptors and the extent of biological effect, irrespective of their concentration and receptor isoform.

According to our quantitative analysis, for all three AHK receptors the correlation between GUS activity and the binding level of both *R-* and *S-*chiral nucleobases was high (correlation coefficients between 0.88 and 0.98). This is consistent with our earlier data on a wide range of various non-chiral BA-derivatives (bases) [26]. This means that the level of hormonal activity of cytokinin nucleobases is generally directly dependent on their affinity for the receptors.

As regards chiral ribonucleosides, the binding of *R*-ribosides at the lower (1 µM) concentration to the AHK receptors did not exceed ~11%, i.e., drastically dropped to a very weak/zero level compared to cognate nucleobases. The only exception among 12 samples was compound **13a,** whose binding to the AHK3 receptor was rather moderate (about 27%, Table 4). However, this exceptional result had no support in the gene expression data (Figure 1 and Appendix A), so its relevance is still ambiguous. When the ligand concentration was raised to 50 µM, an increase in *R*-ribonucleoside binding was observed in almost all variants, especially noticeable for the AHK3 receptor. In addition, the CRE1/AHK4 receptor bound compounds **13a** and **14a** at the levels of ~40% and ~64%, respectively. Surprisingly, such predominance in the binding (Table 4) was not mirrored by the GUS activity biotest where most of the *R*-ribonucleosides elicited an equally active response, at 50 µM, with any of AHK receptors (Table 2).

As for the *S*-ribonucleosides, they did not substantially bind to the AHK2 and CRE1/AHK4 receptors at both concentrations, at neither 1 nor 50 µM. Although there was no specific binding of these compounds at a concentration of 1 µM to the AHK3 receptor as well, one *S*-riboside (**13b**) turned out to moderately (at the level of ~41% of TB) bind to it at 50 µM (Table 4). However, again, this exclusive result poorly correlated with data on gene expression studies which showed that all *S*-ribosides, regardless their affinities for AHK3 receptor, exhibited medium and high activities in the GUS-based transcriptional assay (Table 2).

In contrast to free bases, *R*- and *S*-ribonucleosides exhibited high cytokinin activity in the biotest when taken at very high (50 µM) concentration. Although the latter greatly exceeded physiological concentrations of phytohormones, this phenomenon required a rational explanation. It is conceivable that some ribonucleosides can be partially converted into free bases with higher or lower efficiency during the procedure of in planta biotest. From the total amount of ribosides, most likely only a small fraction is cleaved, so the derived free base concentration might be close to 1 µM, at which point the same nucleobases were applied in the independent experiments. If so, there should be a noticeable correlation between the patterns for 50 µM ribosides and 1 µM bases, but only for those belonging to the same chirality, i.e., *R*-riboside (50 µM) effects should correlate with *R*-base (1 µM) effects, but not with *S*-base (1 µM) effects, and vice versa. To test this hypothesis, a correlation analysis using GUS activity data (Table 2 and Appendix A) was performed. For that the correlation coefficients (CCs) between activities of ribosides (50 µM) and nucleobases (1 µM) were calculated for all receptors, i.e., each 12-point series (Appendix A) rendering results quite relevant.

Indeed, this assumption turned out to be the case: the CCs between *R*-ribosides and *R*-bases, and between *S*-ribosides and *S*-bases were substantial, namely about 0.6 and 0.7, respectively, whereas the cross CCs were close to 0, indicating no correlation. These preferential correlations between activities of *R*- or *S*-nucleobases (1 µM) and cognate ribonucleosides (50 µM) explained well the apparent activities of ribosides taken at high concentration, by their partial decomposition in planta, bearing in mind that ribosides themselves are inactive.

It is noteworthy that compounds **12b** and **16b**, which showed anticytokinin properties in the GUS activity biotest with the AHK2 and CRE1/AHK4 receptors, hardly bound to the cytokinin-binding site of these receptors even at a concentration of 50 µM (Table 4). Similar behavior was noticed for Ado^BOM^ (Appendix A). Ado^BOM^ has been described earlier as a true anticytokinin (BOMA) capable of specifically binding to the CRE1/AHK4 receptor [31]. However, that result was obtained in binding assays based on transgenic *E. coli* expressing individual Arabidopsis receptors. Later, it was shown that the use of such a heterologous test system to analyze the ligand–receptor interaction may provide false-positive results when using cytokinin ribosides [34].

In the present study, we tested the level of Ado^BOM^ (and its free base Ade^BOM^) binding to all three Arabidopsis receptors in an advanced system based on plant microsomes (Appendix A). In this plant-derived binding assay, Ado^BOM^ showed almost no ability to bind specifically to cytokinin receptors even at a concentration 50 µM. This means that Ado^BOM^, similar to compounds **12b** and **16b**, is not a competitive anticytokinin–receptor antagonist.

The lack of detectable binding of the new anticytokinins to the receptors may cast doubt on their action at the receptor level in general. One can imagine, for example, their negative effect at the level of post-receptor signal transduction in a GUS-based expression biotest. However, we still believe that our anticytokinins act directly on the receptors. First, anticytokinins are receptor-specific, which in our biotests, differing only in the isoform of the active cytokinin receptor, is most easily explained by a direct effect on the receptor. Second, unlike the single receptor in our biotest, the signal transduction system includes many redundant components, making it improbable that this system would be blocked by any one type of molecule.

After obtaining new data on the lack of Ado^BOM^ binding to AHK receptors, we additionally tested Ado^BOM^ and Ade^BOM^ for anticytokinin properties in a biotest with seedlings of Arabidopsis receptor mutants (see Section 4). Again, our data were consistent with the former results [31] that Ado^BOM^ significantly suppressed CRE1/AHK4 activation by cytokinin BA. Moreover, our current studies showed that Ado^BOM^ also caused a similar effect when interacting with the AHK2 receptor, the latter being not studied in the referred work. Similarly to the bases **5b** and **8b,** which are cognate for ribosides **12b** and **16b**, respectively, the nucleobase Ade^BOM^ lost the anticytokinin properties inherent to cognate ribonucleoside Ado^BOM^ (Appendix A).

Thus, we showed that ribonucleosides **12b**, **16b**, and Ado^BOM^ exhibited a significant anticytokinin effect, suppressing by 30–50% the activation of AHK2 and CRE1/AHK4 receptors with BA in biotests. For the canonic receptor antagonists, we would expect a partial inhibition of ^3^H-iP specific binding to the same receptors by a similar extent, namely 30–50%. However, the detected inhibition of ^3^H-iP binding to the mentioned receptors by 50 µM **12b** and **16b** did not exceed 12–13% (mean value 7.3%). The nature of this discrepancy remains unclear, and the mode of inhibition resembles that of non-competitive anticytokinin S-4893, which was supposed to interact with the cytokinin receptor at a site different from the ligand-binding one [30]. This additional site was characterized by a relatively low affinity for the reported anticytokinin. A similar trait, namely low affinity for the receptors, was typical for other known anticytokinins, including Ado^BOM^ [31]. Recently, some evidence has been published arguing for the existence of the additional binding site on the surface of the dimeric sensory module of the CRE1/AHK4 receptor [46]. According to the authors, this site recognizes some urea derivatives with adjuvant properties, functioning as allosteric regulator of the cytokinin receptor activity.

However, an additional regulatory site can be located on any receptor module, not necessarily on a hormone-sensing one. Previously it has been found that the isolated C-part of the CRE1/AHK4 receptor was able to bind labeled cytokinin to a very small, but significant, extent [47] (in the mentioned case it was ^3^H-tZ). This may show the existence of a low/medium affinity site for some cytokinins within the non-sensory part of the CRE1/AHK4 receptor. Taking into account that the catalytic module of the receptor contains the binding site for ATP, which represents an adenine derivative, we may speculate that some of the chiral cytokinins, being also adenine derivatives, can compete, especially at high concentration, with ATP for the catalytic module binding. This can form a short but effective negative feedback loop, based solely on the properties of the single protein–cytokinin receptor. Under these assumptions, it can be expected that low cytokinin concentrations will freely stimulate cytokinin receptors, whereas high concentrations along with stimulation will down-regulate the activity of the same receptor, competing with ATP for the ATP-binding site in the catalytic module.

## 3. Molecular Modeling and Docking

### 3.1. In Silico Study of the Interaction of Chiral BA Derivatives with Cytokinin Receptors

To shed more light on the physicochemical basis for the dramatic difference in enantiomer recognition by cytokinin receptors, we accomplished an in silico study using molecular modeling and current bioinformatics tools. The X-ray crystal structure of the sensory module of the Arabidopsis CRE1/AHK4 receptor [48] served as the main framework for modeling and docking. At first, complexes of AHK receptors with *R*- (**5a**–**8a**) and *S*- (**5b**–**8b**) enantiomer bases were obtained by flexible molecular docking, for a total of 27 complexes including control receptor–BA complexes.

According to docking results, a significant difference in *R*- or *S*-isomer binding to receptors AHK2 and CRE1/AHK4 (and a reduction of this difference in AHK3) may be associated primarily with variable residue position located in the ligand-binding site of the PAS domain. This position is occupied by Ile391 in AHK2, Val254 in AHK3 and Leu274 in CRE1/AHK4. In the *S*-isomers, their chiral group is directed precisely towards the indicated residues, forming hydrophobic alkyl–alkyl interactions (according to the extended classification of interactions in the Discovery Studio Visualizer) (Appendix A). In AHK2 and CRE1/AHK4, which have residues with a bulkier side chain (Ile and Leu), the position of the *S*-isomers was considerably shifted compared to the position of the *R*-isomers and control BA molecule. Such shifting obviously reduced the affinity of the *S*-enatiomers for the receptors. In AHK3, the residue in this position (Val) has the smallest side chain, resulting in much less deviation in the *S*-isomer position (Figure 3). Although the hydrophobic amino acids at the indicated position could play a critical role in differential recognition of BA-derived *R*- and *S*-enantiomers, the participation of some other amino acids within binding site cannot be excluded as well.

Evidently, the difference in the binding of the chiral BA derivatives, carrying various substituents at *C*2, to AHK receptors cannot be only explained by the difference of size of these substitutes. We calculated van der Waals volumes of the entire nucleobases and single radicals at *C*2 (Appendix A). The volume of the amino group, which seemed bulky and thus prevented the compound binding to the sensory module of the receptor, turned out to be smaller than that of the chlorine substituent. At the same time, the cytokinin activity and affinity to receptors of compounds bearing an amino group at *C*2 was lower compared to an analogous compound with chlorine substitution. Hence, the difference in the binding apparently arises, at least partially, from the different electrostatic potentials of the ligand molecular surface close to the substituent position (Figure 4). Fluorine- or chlorine-substituents render the charge more negative; conversely, an amino group will render it positive. Such a difference in physicochemical properties may account for a difference in anticytokinin activity between the *S*-ribonucleoside pair **12b**/**16b** (no halogen at *C*2 position, active anticytokinins) and pair **13b**/1**4b** (halogens at *C*2 position, active cytokinins). It is likely that with some other *C*2 substituents, negatively charged and limited in volume, the series of *S*-nucleobases selectively interacting with AHK3 could be extended.

Assessment of binding energies of the ligand–receptor complexes (single minimized structures) by PBS method in the Yasara program showed some correspondence between computed and experimental data. Overall *S*-bases appeared to bind weaker than *R*-bases to the receptors, and the strongest *S*-bases binding was attributed to AHK3 (Appendix A).

### 3.2. Difference between R- and S-Derivative Binding in Molecular Dynamics Simulation

We performed a 50 ns molecular dynamics (MD) simulation for 10 selected structures: complexes of compounds **6a**, **6b**, **7a**, **7b** and BA with AHK2 and AHK3 receptors. Calculation of ligand binding energies based on a MDtrajectory also showed consistency with the experimental results. The binding energy of **6b** by the AHK2 receptor was much lower than that of **6a**. However, in the case of the AHK3 receptor, such a difference was greatly reduced (Figure 5), while each of the enantiomers binds to AHK3 more strongly than to AHK2.

The latter was also true for compounds **7a** and **7b** which have chlorine substitution in adenine heterocycle (Appendix A). Control cytokinin BA showed weaker binding to the AHK3 than to AHK2 receptor (Appendix A), in agreement with the published experimental data [34]. The correlation coefficient between the two data series (calculated and experimental) was quite high (0.864), which gives prognostic value to the applied simulation algorithm. It is noteworthy that the AHK2–**6b** complex was distinguished by much lower solvation energy than other protein**–**ligand pairs studied here by the MD method (Appendix A).

Thus, in silico modeling provided reasonable explanations for some observed features of the interaction of chiral BA derivatives with Arabidopsis cytokinin receptors. Although the proposed model is still tentative, it may stimulate new investigations in this area.

## 4. Materials and Methods

### 4.1. Chemical Synthesis

#### 4.1.1. General

The solvents and materials were reagent grade and were used without additional purification. Column chromatography was performed on silica gel (Kieselgel 60 Merck, Darmstadt, Germany, 0.063–0.200 mm). TLC was performed on an Alugram SIL G/UV254 (Macherey-Nagel, Düren, Germany) with UV visualization. The ^1^H and ^13^C (with complete proton decoupling) NMR spectra were recorded on Bruker AVANCE II 300 (Karlsruhe, Germany) instrument at 303 K. The ^1^H-NMR-spectra were recorded at 300.1 MHz, the ^13^C-NMR-spectra at 75.5 MHz. The chemical shifts in ppm were measured relative to the residual solvent signals as internal standards (DMSO-*d*_6_, 1H: 2.5 ppm, ^13^C: 39.5 ppm; CD_3_OD,^13^C: 49.0 ppm). Spin–spin coupling constants (*J*) are given in Hz.

The high-resolution mass spectra (HRMS) were registered on a Bruker micrOTOF II instrument using electrospray ionization (ESI) [49]. The measurements were taken in a positive ion mode (interface capillary voltage e 4500 V); mass range from *m*/*z* 50 to *m*/*z* 3000 Da; internal calibration was undertaken with electrospray calibrant solution (Fluka). A syringe injection was used for solutions in acetonitrile: water mixture, 50/50 vol. % (flow rate 3 mL/min). Nitrogen was applied as a dry gas; interface temperature was set at 180 °C.

We purchased 6-Chloropurine (**1**) (CAS 87-42-3) and 2-amino-6-chloropurine (**4**) (CAS 10310-21-1) from Sigma-Aldrich (sigmaaldrich.com (accessed on 14 January 2019)). 2,6-Dichloropurine (**3**) and 2-fluoro-6-chloropurine (**2**) and 1-(β-D-ribofuranosyl)-2-amino-6-chloropurine (**15**) were obtained according to the literature procedures, described in [50,51,52]. Purine nucleoside phosphorylase *E. coli* (EC 2.4.2.1, CAS 9030-21-1) was purchased from Sigma-Aldrich (sigmaaldrich.com (accessed on 14 January 2019)).

All synthetic procedures and spectral data of the obtained compounds are given in detail in the Appendix A.

#### 4.1.2. Synthesis of *N*^6^-Alkyladenines

Various 6-chlorosubstituted purines (**1**–**4**) were introduced into amination reaction with *R*- and *S*-(α-methylbenzyl)amine in the presence of DIPEA (Figure 1, Results and Discussion). The presence of substituents at position 2 of purine strongly influenced the reaction rate and determined the choice of reaction conditions. Amination of 6-chloropurine (**1**) at 60 °C proceeded slower than for 2-fluoro-6-chloropurine (**2**) and 2-chloro-6-chloropurine (**3**), according to TLC this was probably due to steric hindrance of the secondary amino group in *R-* and *S*-(α-methylbenzyl)amine. Elevation of temperature to 110 °C afforded faster achievement of full conversion of **1** to *N*^6^-monosubstituted products **5a** and **5b** after 4.5 h of reaction mixture heating. The products **5a** and **5b** were isolated with 37% and 56% yield after column chromatography and recrystallization from *R-* and *S*-(α-methylbenzyl)amine, DIPEA and their hydrochloric salts. An electron-donating aminogroup at position 2 of 2-amino-6-chloropurine (**4**) substantially decreased its reactivity to nucleophilic substitution with sterically hindered secondary amines. As a result, amination of **4** with *R-* and *S*-(α-methylbenzyl)amine in the presence of DIPEA proceeded slower than for **1**. Refluxing of the reaction mixture at 120 °C afforded full conversion of compound **4** to *N*^6^-monosubstituted products **8a** and **8b** after 6.5 h according to TLC. Compounds **8a** and **8b** were isolated with 20% and 38% yields, respectively, after reaction mixture work-up. However, the electron-withdrawing substituents (Y = F, Cl; Table 1) at position 2 of 2-fluoro-6-chloropurine (**2**) and 2-chloro-6-chloropurine (**3**) favored mild reaction conditions. In the case of amination of **2**, partial substitution occurred at position 2 of the purine even at 60 °C, decreasing yields of the *N*^6^-monosubstituted products **6a** and **6b** to 34% and 25% respectively. For more details see synthetic procedures in Appendix A.

#### 4.1.3. Synthesis of Ribonucleosides

An efficient approach based on activation of position 6 of hypoxanthine and guanine in ribonucleosides by benzotriazole-1-yl-oxy-tris-(dimethylamino)-phosphonium hexafluorophosphate (BOP) was applied for synthesis of chiral *N*^6^-substituted adenosine (**12ab**) (Figure 2, route i, Results and Discussion) and guanosine (**16a** and **17b**) derivatives (Figure 5, route i, Results and Discussion). The significance of this reaction is explained by the activation of the sixth position of inosine (guanosine), followed by the possibility of replacing the benzotriazole residue with the methylbenzylamine residue under mild conditions (room temperature and below) without the formation of by-products. Reactions with BOP proceeded either through the formation of an activated complex of the hexamethylphosphonium salt, or through the formation of benzotriazol-1-yl-substituted intermediate, which readily interacts with various nucleophiles to form corresponding substitution products [53,54,55,56,57]. According to the literature and our data, nucleophilic substitution in the presence of BOP can proceed both for *O*-deprotected and *O*-protected hypoxantine and guanine ribosides [58,59]. BOP-catalyzed reactions give higher yields when using *O*-protected ribosides, with reaction conditions well-studied for fully-*O*-silylated or fully-*O*-acetylated ribosides [59]. In the scope of the present work, we proposed fully *O*-isobutyroyl protected inosine (**9**) and guanosine (**17**–) as alternative substrates for BOP-activation. The choice of the isobutyroyl protective group was due to its higher stability under treatment with amines compared to commonly used acetyl group and its easy removal under mild conditions [60]. In our case the reaction of **9** with BOP in the presence of DIPEA proceeded with the formation of an activated *O*^6^-(benzotriazol-1-yl)-2′,3′,5′-tri-*O*-isobutyroyl inosine (**10**), which was isolated with near quantitative yield (Figure 2). In the case of unprotected inosine and guanosine the reaction gave significantly lower yields of triazol-activated products (26% and 35% respectively). The benzotriazol-activated derivative **10** was then introduced into condensation with *R*- and *S*-α-methylbenzylamine to form derivatives **11ab** in 84% and 79% yield, respectively. Isobutyroyl protective groups were removed by treatment of **11ab** with 4M methylamine solution in ethanol with the formation of **12ab** in 82% and 94% yields, respectively (Figure 2, Results and Discussion).

We obtained 2-Fluoro- (**13ab**) and 2-chloro- (**14ab**) substituted adenosines by glycosylation of the corresponding *N*^6^-(α-methylbenzyl)-2-fluoro- (**6ab**) and 2-chloro- (**7ab**) substituted adenine derivatives (Figure 3, Results and Discussion); the presence of bulky methylbenzyl substituent at position 6 favored the course of the reaction towards the formation of *N*9-glycosylated products [61,62,63,64]. The further step of deblocking the acetyl protecting groups was carried out in 2M NH_3_/MeOH solution at −7 °C in a freezer to prevent unfavorable nucleophilic substitution at *C*2 position of 2-fluoro- and 2-chloropurine residue. Only under these conditions did the removal of acetyl groups proceed selectively without the formation of 2-amino by-products, while even at 0 °C the deacylation reaction was accompanied by the formation of 2-methyl-substituted by-products (see Appendix A).

We obtained 2-Amino-substituted adenosine derivatives with chiral *N*^6^-substituents **16ab** by direct amination of unprotected 2-amino-6-chloropurine riboside (**15**) by *R-* and *S*-(α-methylbenzyl)amine (Figure 4, Results and Discussion). While, upon heating of compound **15** with *R*- and *S*-(α-methylbenzyl)amine, full conversion of **15** was not observed according to TLC and darkening of the reaction mixture did occur, BOP activation was a good alternate procedure affording the formation of products in mild conditions with high conversion of **15**, analogous to the preparation of **12ab**.

To obtain *R*- and *S*-isomers of 2-amino,*N*^6^-(α-methylbenzyl)adenosine, two synthetic pathways involving two various benzotriazole-activated guanosine derivatives **18** and **21** were performed (Figure 5, Results and discussion). At first, 2′,3′,5′-tri-*O*-isobutyroylguanosine (**17**) and *N*^2^-isobutyroyl-2′,3′,5′-tri-*O*-isobutyroyl guanosine (**20**) were treated with BOP reagent to form the corresponding benzotriazole-activated nucleosides **18** and **21** in high yields, which were then introduced into condensation with *R-* and *S*-α-methylbenzylamines with the formation of **19** and **22ab**, respectively. Deacylation of **19** afforded pure product **16a** in good overall yield (76%), while, in the case of deacylation of tetraisobutyroyl-protected **22a,** the formation of partially protected product **23a** was observed. The presence of *N*^2^-isobutyroyl group was confirmed by ^1^H-NMR-spectroscopy in DMSO-*d_6_*, where one signal of isobutyric C-*H* proton in a form of multiplet at 2.9 ppm and two signals of CH_3_ groups in a form of doublets at 1.1 and 1.0 ppm were present in a low-field region (see Appendix A). Compound **23a** was stable upon storage in 4M MeNH_2_/EtOH for 4 days at ambient temperature and even upon heating of the reaction mixture.

It thus can be concluded from the obtained results that introduction of chiral secondary amine group by substitution at C-6 of *O*^6^-benzotriazol-activated guanine derivative proceeds more efficiently than for 6-chloro-activated guanine and that *O*^6^-benotriazolyl-2′,3′,5′-tri-*O*-isobutyroylguanosine (**19**) is a more preferable candidate for amination than *O*^6^-benotriazolyl-*N*^2^-isobutyroyl-2′,3′,5′-tri-*O*-isobutyroylguanosine (**20**) due to high stability of *N*^2^-isobutyric group under deprotection conditions. For more details see synthetic procedures in Appendix A.

#### 4.1.4. Synthesis of Ado^BOM^ and Ade^BOM^

Ado^BOM^ was synthesized by regioselective alkylation of *N*^6^-acetyl-2′,3′,5′-tri-*O*-acetyladenosine (**24**) with benzyloxymethyl chloride in the presence of DBU with subsequent deacylation in the presence of propylamine in methanol, according to the method reported earlier (Figure 6, routes i and ii, Results and Discussion) [65]. Ado^BOM^ can then be converted into Ade^BOM^ using an effective approach based on the enzymatic arsenolysis of purine nucleosides with purine nucleoside phosphorylase (PNP) (Figure 6, route iii). PNP-catalyzed arsenolysis in the presence of potassium dihydroorthoarsenate (KH_2_AsO_4_) is based on the cleavage of *N*-glycosidic bond of ribonucleoside with the formation of a purine base and highly labile α-*D*-ribofuranose-1-arsenate (Rib-1-As), which is irreversibly hydrolyzed [60,66]. For more details see synthetic procedures in Appendix A.

### 4.2. Plant-Based Methods

#### 4.2.1. Cytokinin and Anticytokinin Activity Assay

(Anti)cytokinin activity of the compounds was evaluated in an assay system based on *Arabidopsis thaliana* (L.) Heynh. seedlings. We used double mutants of Arabidopsis with only one of three cytokinin receptors (AHK2, AHK3, or CRE1/AHK4) kept active [12]. The seeds of the mutants were kindly provided by Prof. T. Schmülling (Freie Universität Berlin, Germany). All plants harbored the reporter *GUS* gene driven by cytokinin-dependent promoter *P_ARR5_* [43]. Four-to-five-day old Arabidopsis seedlings were incubated for 16 h in solutions of the compounds [67].

The stock solutions of tested compounds in 100% DMSO at a concentration of 0.1 M were diluted to the desired concentrations with distilled water. In case of cytokinin activity determination final concentration of compounds was 10^−6^ M. The cytokinin effect of each separate compound was quantified by determination of the level of GUS activity reflecting the intensity of the *P_ARR5_:GUS* expression [68].

To determine the anticytokinin activity of the compounds, we compared the activity of BA at a concentration of 10^−7^ M and a mixture of BA at the same concentration with the compound under test at a concentration of 50 μM. This excessive concentration of potential anticytokinin over the control cytokinin was adopted from studies of currently known anticytokinins [28,29,31]. If the activity of the mixture was lower than that of the control, this was considered a manifestation of the anticytokinin effect of the compound.

#### 4.2.2. Histochemical Determination of GUS Activity

Four-day-old mutant Arabidopsis seedlings with only one of three cytokinin receptors (AHK2, AHK3, or CRE1/AHK4) were incubated for 16 h in distilled water (negative control), solutions of tested compounds at a concentration of 10^−6^ M (assayed probes) or BA (positive control) at the same concentration. A substrate solution was prepared. The stock solution of X-Gluc (5-bromo-4-chloro-3-indolyl-β-D-glucuronide) in 100% DMSO at a concentration of 0.1 M was diluted to the concentration 1 mM with 50 mM sodium phosphate buffer (Na_2_HPO_4_ and NaH_2_PO_4_ solution in distilled water, pH 7). We added 0.1% Triton X-100 and 1 mM EDTA. After incubation, seedlings were washed with distilled water and placed in a solution of X-Gluc that completely covered them. Plants were incubated in this solution for another 16 h at 37 °C. The reaction was stopped by washing the samples with distilled water. Then the seedlings were extracted with 70% ethanol [68]. The photos of the seedlings were taken with a microscope Axio Imager D1 using the appropriate software.

#### 4.2.3. Expression of Cytokinin Receptor Genes and Membrane Fraction Isolation

Transformation of tobacco (*Nicotiana benthamiana* Domin) leaves was carried out according to [69]; eight-week-old plants were used. Infiltration was made with a mixture of *Agrobacterium tumefaciens* clones harboring recombinant plasmids: the clone carrying one of the Arabidopsis cytokinin receptor genes fused with *GFP* (OD600~0.035) was mixed with the clone carrying the *p19* gene (OD600~0.05) [70]. A transient expression of target genes in tobacco was detected using a fluorescence microscope AxioImager Z2 (Carl Zeiss Microscopy GmbH, Oberkochen, Germany). Leaves showing strong expression of receptors were used for membrane isolation [35]. They were ground with a mortar and pestle in homogenization buffer (100 mM Tris-HCl pH 7.6, 10 mM Na_2_-EDTA, 1 mM dithiothreitol, 1 mM phenylmethylsulfonyl fluoride). The obtained homogenate was filtered through Miracloth (Calbiochem^®^, San Diego, CA, USA). The filtrate was centrifuged at 5000 *g* for 10 min, removing cell debris and large organelles. Then the supernatant was centrifuged at 18,500× *g* for 20 min. The last supernatant was removed and the membrane pellet was carefully resuspended in 50 mM KCl + 10% glycerol solution.

#### 4.2.4. Cytokinin Binding Assay

Direct binding of compounds to Arabidopsis cytokinin receptors was assessed by the radioligand method based on the receptor ability to bind cognate ligands. Binding was performed according to the method described in [34] with modifications [35]. Tritium-labeled isopentenyladenine (^3^H-iP, 2 pmol per sample, specific activity 17.4 Ci/mmol [71]) was used as a labeled ligand. Total (TB) and nonspecific (NS) binding were determined by measuring binding in samples containing only ^3^H-iP or ^3^H-iP together with large (500-fold) excess of unlabeled iP, respectively. Specific binding (SB) was defined as the difference between TB and NS [34]. Unlabeled ligands were used at a concentration of 1 µM and 50 µM. All samples with membranes obtained by different methods were normalized in binding assays in terms of protein content.

#### 4.2.5. Statistical Analysis

The experiments were carried out in 2–3 biological replications. In each experiment, data are means of two independent determinations with standard errors (SE). Correlation coefficients were determined according to Pearson’s algorithm.

### 4.3. Computational Methods

For molecular docking, models of sensory modules of AHK cytokinin receptors obtained earlier [34,72] were used. The structures of chiral isomers of *N*^6^-benzyladenine derivatives were obtained using the MolView service [73]. GLmol was used as render engine. Docking was performed with VINA [74] using default parameters. The setup was completed with the YASARA molecular modeling program [75]. Minimum of ligand RMSD was set to 5.0 Å. Positions with the best energy score were selected for further use, provided that the position of the ligand was similar to that in the crystal structure of CRE1/AHK4 (PDB ID: 3T4K).

The binding energies for the single minimized structures were calculated in YASARA [75] using a BindEnergy_WithSolvation macro with PBS solvation energy method, AMBER14 force field, and boundary periodic cell with extension parameter set to 20. Minimization procedures were performed using an em_run macro with default settings. Van der Waals volumes were calculated using MoloVol 1.0.0 [76]. Small probe radius was set to 1.2 Å. Grid resolution was 0.2 Å and optimization depth was set to 4.

Molecular dynamics simulations were performed with YASARA Structure software [75]. The setup included an optimization of the hydrogen bonding network [77] to increase the solute stability, and a pKa prediction to fine tune the protonation states of protein residues at the chosen pH of 7.4 [78]. NaCl ions were added simulating a physiological concentration of 0.9%, with an excess of either Na or Cl to neutralize the cell. Simulation temperature was 298 K. Water density was 0.997 g/mL. Shape of the simulation cell was set to Cuboid. Extension of the cell on each side around the solute was 10 Å. Cell boundary was periodic. Drift correction was turned on to keep the solute from diffusing around and crossing periodic boundaries. After steepest descent and simulated annealing minimizations to remove clashes, the simulation was run for 50 nanoseconds using the AMBER14 force field [79] for the solute, GAFF2 [80] and AM1BCC [81] for ligands and TIP3P for water. The cutoff was 8 Å for van der Waals forces (the default used by AMBER [82]), no cutoff was applied to electrostatic forces (using the particle mesh Ewald algorithm, [83]). The equations of motions were integrated with a multiple time step of 2.5 fs (md_runfast protocol) for bonded interactions and 5.0 fs for non-bonded interactions at a temperature of 298 K and a pressure of 1 atm (NPT ensemble) using algorithms described in detail previously [84].

Trajectories were analyzed using md_analyze and md_analyze bind energy macros in YASARA Structure. Binding energies were analyzed using both BoundaryFast and Poisson–Boltzmann (PBS) methods. The last one is equal to ‘MM/PBSA’, just without the entropy term from normal mode analysis. Temperature was 298 K and force field was AMBER14.

In addition to the YASARA tools, protein–ligand interactions were analyzed in the LigPlot [85] implemented in PDBsum web-application [86] and Discovery Studio Visualizer [87]. Calculation of the electrostatic potential and mapping on the molecular surface was carried out in Chimera 1.14 [88]. This program was also used to visualize models.

## 5. Conclusions

Here we described the synthesis and the study of a series of new analogues of natural cytokinin BA, including both nucleobases and ribonucleosides, all containing a chiral substituent in the *N*^6^ side chain as well as different substituents at the *C*2 position of the purine heterocycle. The resulting compounds were examined for cytokinin and anticytokinin activity as well as binding to the Arabidopsis cytokinin receptors AHK2, AHK3, and CRE1/AHK4. BA derivatives containing the *R-*chiral fragment in their structure were shown to exhibit pronounced cytokinin properties in interaction at rather moderate concentration (1 µM) with all three receptors. In contrast, the *S-*isomers, especially the *S*-nucleobases, exhibited significant activity only with the AHK3 receptor. Due to this receptor selectivity of *S*-nucleobases, several new synthetic receptor-specific cytokinins (*S*-**MBA** (*N*^6^-(α-methylbenzyl)adenine)), *S*-**FMBA** (2-fluoro,*N*^6^-(α-methylbenzyl) adenine), *S***-CMBA** (2-chloro,*N*^6^-(α-methylbenzyl)adenine)) were uncovered which interacted at close to physiological concentration almost exclusively with the AHK3 receptor.

Nucleobases and ribonucleosides lacking significant cytokinin activity at concentrations up to 50 µM were checked for the putative anticytokinin activity. As a result, two new compounds, *S*-ribonucleosides *N*^6^-((*S*)-α-methylbenzyl)adenosine and 2-amino,*N*^6^-((*S*)-α-methylbenzyl)adenosine, were found to exhibit a marked anticytokinin effect against AHK2 and CRE1/AHK4 receptors. At the same time, these compounds were shown not to be competitive anticytokinins since they apparently had no high affinity for cytokinin receptors. A close compound Ado^BOM^, long considered a competitive receptor antagonist, was shown to have similar characteristics. Despite this, its anticytokinin activity was confirmed in a CRE1/AHK4 receptor bioassay and even extended to the AHK2 receptor.

Collectively, this study of BA-derived *R*- and *S*-enantiomers, carried out for the first time, confirmed the patterns established in our previous works with numerous non-chiral analogs of cytokinins [26]. These are: (i) the blocking effect of *N*9 ribosylation on the cytokinin activity of adenine derivatives; (ii) the strong correlation between cytokinin receptor binding of a particular compound with its hormonal activity; (iii) the enhancement of the hormonal potential of the adenine derivative by a halogen (predominantly F) substituent at the purine position *C*2; and (iv) the close similarity in ligand preference between the Arabidopsis receptors AHK2 and CRE1/AHK4. However, the main outcome of this work is the clear demonstration of the important role that the ligand chirality can play in the cytokinin–receptor interaction, as well as the discovery of the first receptor-specific cytokinins and new anticytokinins among the *S*-enantiomers of BA-derivatives.

Along with in silico modeling and docking, this study provides new insight in the mode of AHK receptor functioning in the perception of the cytokinin and anticytokinin signals. The found ligands with unique properties can be further explored for scientific and practical purposes, while the raised questions open prospects for future investigations.

## Data Availability

Not applicable.

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
