# Peer review of "In Planta, In Vitro and In Silico Studies of Chiral N6-Benzyladenine Derivatives: Discovery of Receptor-Specific S-Enantiomers with Cytokinin or Anticytokinin Activities"

_ijms, 2022, doi:10.3390/ijms231911334_

Round 1

Reviewer 1 Report

The manuscript is well written and the experimental work is well described. However, the novelty and importance of the work should be emphasised.  The abstract should be strictly describing the results and should not include introduction information. In the key words I find that "chiral compound" is a too general word to include in key words. Similar with R- and S-enantiomers,  too general, the enantiomers should be named. The heading 2.1 in line 96 is also too general, "Chemistry" can be all of chemsitry, and here it is not. Be precise. In page 5, line 136 the authors claim that R- and S-enantiomers cannot be distinguished in NMR, this is of course common knowledge, and should be written in another way. Be precise. In page 5, Table 1, the table text is not informative, and should be improved. In page 10, the n/d - not determined should have an asterisk in the table and also below the table. the same in table 4, page 11. Page 21, conclusion line 776: Delete "this work resulted in..", just descibe the conclusion of the results. Page 22, line 797: Delete "Taken together". And it should not be necessary to stat "for the first time" since if this was reported before, it should not be published once more. Line 804: Delete "explicit demonstration" .

Author Response

Author's Reply to the Review Report (Reviewer 1)

Comments and Suggestions for Authors

Point 1. The manuscript is well written and the experimental work is well described. However, the novelty and importance of the work should be emphasised.  The abstract should be strictly describing the results and should not include introduction information.

Response 1. We thank the Reviewer for the positive evaluation of our work and useful comments that can certainly improve our paper. We agree with the Reviewer that the abstract should primarily contain the results, therefore in the total abstract only one sentence serves to introduce the main actors, i.e. plant hormones cytokinins and their receptors, as well as the main issue concerning cytokinin practical application that should be addressed. Without this brief explanation most readers will not understand the main purpose and importance of the undertaken study. 

Point 2. In the key words I find that "chiral compound" is a too general word to include in key words. Similar with R- and S-enantiomers,  too general, the enantiomers should be named.

Response 2. We would be happy to be precise and specify our chiral compounds and enantiomers, but, unfortunately, there are yet no commonly accepted names for them. And to list their 8 long chemical names as key words is not possible. By contrast, the terms "chirality" and "enantiomer" are widely known, and our article would be easily found in the literature database by combining key words "cytokinin" and "chirality" or "enantiomer".

Point 3. The heading 2.1 in line 96 is also too general, "Chemistry" can be all of chemsitry, and here it is not. Be precise.

Response 3.  We have replaced the heading 2.1. with more precise "Chemical Synthesis"

Point 4. In page 5, line 136 the authors claim that R- and S-enantiomers cannot be distinguished in NMR, this is of course common knowledge, and should be written in another way. Be precise.

Response 4. We agree that this fact regarding NMR may be obvious to chemists. However, the article is aimed at a wide audience, for the majority of which such a statement is not obvious. And this sentence was written in such a way as to explain in as much detail and intelligibly as possible why certain methods were used.

Point 5. In page 5, Table 1, the table text is not informative, and should be improved.

Response 5. The table text was supplemented with more details and became more informative.

Point 6. In page 10, the n/d - not determined should have an asterisk in the table and also below the table. the same in table 4, page 11.

Response 6. We added the asterisks to both tables.

Point 7. Page 21, conclusion line 776: Delete "this work resulted in..", just descibe the conclusion of the results. Page 22, line 797: Delete "Taken together". And it should not be necessary to stat "for the first time" since if this was reported before, it should not be published once more. Line 804: Delete "explicit demonstration" .

Response 7. The Reviewer 1 suggestions were implemented in the ms text, either by deleting or transforming mentioned expressions.

Author's Reply to the Review Report (Reviewer 1)

Comments and Suggestions for Authors

Point 1. The manuscript is well written and the experimental work is well described. However, the novelty and importance of the work should be emphasised.  The abstract should be strictly describing the results and should not include introduction information.

Response 1. We thank the Reviewer for the positive evaluation of our work and useful comments that can certainly improve our paper. We agree with the Reviewer that the abstract should primarily contain the results, therefore in the total abstract only one sentence serves to introduce the main actors, i.e. plant hormones cytokinins and their receptors, as well as the main issue concerning cytokinin practical application that should be addressed. Without this brief explanation most readers will not understand the main purpose and importance of the undertaken study. 

Point 2. In the key words I find that "chiral compound" is a too general word to include in key words. Similar with R- and S-enantiomers,  too general, the enantiomers should be named.

Response 2. We would be happy to be precise and specify our chiral compounds and enantiomers, but, unfortunately, there are yet no commonly accepted names for them. And to list their 8 long chemical names as key words is not possible. By contrast, the terms "chirality" and "enantiomer" are widely known, and our article would be easily found in the literature database by combining key words "cytokinin" and "chirality" or "enantiomer".

Point 3. The heading 2.1 in line 96 is also too general, "Chemistry" can be all of chemsitry, and here it is not. Be precise.

Response 3.  We have replaced the heading 2.1. with more precise "Chemical Synthesis"

Point 4. In page 5, line 136 the authors claim that R- and S-enantiomers cannot be distinguished in NMR, this is of course common knowledge, and should be written in another way. Be precise.

Response 4. We agree that this fact regarding NMR may be obvious to chemists. However, the article is aimed at a wide audience, for the majority of which such a statement is not obvious. And this sentence was written in such a way as to explain in as much detail and intelligibly as possible why certain methods were used.

Point 5. In page 5, Table 1, the table text is not informative, and should be improved.

Response 5. The table text was supplemented with more details and became more informative.

Point 6. In page 10, the n/d - not determined should have an asterisk in the table and also below the table. the same in table 4, page 11.

Response 6. We added the asterisks to both tables.

Point 7. Page 21, conclusion line 776: Delete "this work resulted in..", just descibe the conclusion of the results. Page 22, line 797: Delete "Taken together". And it should not be necessary to stat "for the first time" since if this was reported before, it should not be published once more. Line 804: Delete "explicit demonstration" .

Response 7. The Reviewer 1 suggestions were implemented in the ms text, either by deleting or transforming mentioned expressions.

Reviewer 2 Report

The manuscript entitled “In Planta, In Vitro and In Silico Studies of Chiral N6-Benzyladenine Derivatives: Discovery of Receptor-Specific S-Enantiomers with Cytokinin or Anticytokinin Activities” by Savelieva et al studied a series of chiral N6-benzyladenine derivatives as potential cytokinins or anticytokinins, in which the interaction patterns between individual receptors and specific enantiomers were rationalized by structure analysis and molecular docking. Therefore, their outcomes are strongly relevant with the readership of International Journal of Molecular Medicine. The revised manuscript can be considered for acceptance after the following minor issues are addressed.

1. The Receptor-Specific can be added in the keywords of the manuscript.

2. For the background colors in Table2-4, the author should give the basis of the classification for the data values analysis.

3. The paragraphs of “Enantiomers are isomers whose structures are mirrored to each other” and “In nature, macromolecules are homochiral, since proteins and enzymes... in the introduction can be summarized in a whole, for the same statement of the important role in the selective ligands-enantiomers binding behaviors with different spatial arrangement.

4. The authors demonstrated the difference in the binding apparently arises from the different electrostatic potentials from molecular modeling and docking, how about its interference for the ligand-receptor complexes between S-bases and R-bases.

Author Response

Author's Reply to the Review Report (Reviewer 2)

Comments and Suggestions for Authors

The manuscript entitled “In Planta, In Vitro and In Silico Studies of Chiral N6-Benzyladenine Derivatives: Discovery of Receptor-Specific S-Enantiomers with Cytokinin or Anticytokinin Activities” by Savelieva et al studied a series of chiral N6-benzyladenine derivatives as potential cytokinins or anticytokinins, in which the interaction patterns between individual receptors and specific enantiomers were rationalized by structure analysis and molecular docking. Therefore, their outcomes are strongly relevant with the readership of International Journal of Molecular Medicine. The revised manuscript can be considered for acceptance after the following minor issues are addressed.

We thank the Reviewer for the positive evaluation of our manuscript and its detailed analysis.

  1. The “Receptor-Specific” can be added in the keywords of the manuscript.

Response 1. Accepted.

  1. For the background colors in Table2-4, the author should give the basis of the classification for the data values analysis.

Response 2. Accepted.

  1. The paragraphs of “Enantiomers are isomers whose structures are mirrored to each other” and “In nature, macromolecules are homochiral, since proteins and enzymes...” in the introduction can be summarized in a whole, for the same statement of the important role in the selective ligands-enantiomers binding behaviors with different spatial arrangement.

Response 3. The mentioned paragraphs were modified.

  1. The authors demonstrated the difference in the binding apparently arises from the different electrostatic potentials from molecular modeling and docking, how about its interference for the ligand-receptor complexes between S-bases and R-bases.

Response 4. The effect of the electrostatic potential on the differences in the binding of R- and S-bases is not so obvious. This is due to the fact that the considered chiral isomers have the same set of atoms, provided that the substituent in the C2 position is the same, and the main structural differences relate to the orientation of the methyl group, while the charge distribution on the ligand surface changes insignificantly. On the other hand, a change in the spatial position of the ligand in the binding pocket, caused by differences in the orientation of the methyl group in enantiomers, can affect the complementarity of the electrostatic potentials of the receptor and the compound, due to a simple displacement of molecular surfaces relative to each other. This issue will be discussed in more detail in subsequent publications.

Reviewer 3 Report

Overall:

The manuscript describes the chemical synthesis and evaluation of a series of nucleoside molecules that systematically vary the structure. The significance of the research is nicely communicated (lines 37-84). The innovative aspects of the current study is precisely and succinctly articulated (line 85-94).

The results and discussion of the chemical synthesis is omitted from the results and discussion section and instead included in the materials and methods. For example, although the ribonucleoside derivatives are justified (lines 111-115) in the results and discussion, their synthesis (reagents used, reactions used, yields, etc.) are not described until the materials and methods (lines 588-657). The methods to prepare the compounds are included in the supplementary material. This organization of synthesis information is unusual for chemistry journals. Normally the general description of the reagents is included in the results and discussion , while the detailed procedural information is placed in the materials and methods or exclusively in the supplementary materials.

The chemical synthesis methods are generally well-described and seemingly well-executed. They adhere to the norms in the chemical synthesis field with one exception. Typically 13C NMR are reported to 1 decimal place. (i.e. 77.7 ppm).

The biological results are presented clearly for a general audience including in vitro data and histochemical staining. These studies are done for both cytochine and anti-cytokine activity. The results and tables are numerous. Therefore, it may be beneficial to more clearly articulate the hit compounds in the conclusion section using compound numbers. For example, 2 new compounds are alluded to in lines 789-793 but are not referenced by structure or compound number.
Clarity:
line 73: They do not overlap because of the different spatial arrangement of the substituents around the optically active (chiral) carbon atom.
*The identity of a structure with its mirror image is somewhat unclear by the statement above. Perhaps: "Their structures are non-identical because..." would be more clear.

line 77: In nature, ...
*The statements about the inherent handedness of molecules in nature are generally true but are not wholly accurate. The S enantiomer of cysteine is natural while glycine is achiral. Also L-fucose is a common L-sugar in mammals and many other L configured carbohydrates are common in microorganisms. Some minor qualifying terms are necessary to make the claims wholly true.

Grammar suggestions:
line 19: All compounds contained a methyl group at the α-carbon atom of the benzyl moiety being R- or S-enantiomers.

line 48: Despite their structural similarity, receptors of the same family often have differences in ligand specificity, which seems to be related to their participation in the long-distance root-to-shoot signaling.

Line 102: To modify natural cytokinin N6-benzyladenine (BA) molecule with the α-methylbenzyl
chiral fragment, various 6-chlorosubstituted purines were introduced into the amination reaction with R- and S-(α-methylbenzyl)amine in the presence of Hunig’s base DIPEA
(Scheme 1) as was previously shown for non-chiral purine derivatives [19,20].

Author Response

Author's Reply to the Review Report (Reviewer 3)

Comments and Suggestions for Authors

Overall:

The manuscript describes the chemical synthesis and evaluation of a series of nucleoside molecules that systematically vary the structure. The significance of the research is nicely communicated (lines 37-84). The innovative aspects of the current study is precisely and succinctly articulated (line 85-94).

We thank the Reviewer for the positive evaluation of our work and useful comments.

Point 1. The results and discussion of the chemical synthesis is omitted from the results and discussion section and instead included in the materials and methods. For example, although the ribonucleoside derivatives are justified (lines 111-115) in the results and discussion, their synthesis (reagents used, reactions used, yields, etc.) are not described until the materials and methods (lines 588-657). The methods to prepare the compounds are included in the supplementary material. This organization of synthesis information is unusual for chemistry journals. Normally the general description of the reagents is included in the results and discussion , while the detailed procedural information is placed in the materials and methods or exclusively in the supplementary materials.

Response 1. Our work indeed may be considered a combination of the chemical and biological parts, where chemical part is very important but plays here a preparative role whereas new data arise from biological experiments. Therefore in the Results and Discussion section mostly biological results prevail, and the chemical part is represented in this section only by the essential synthesis schemes without specifying methodical details which are included in the Materials and Methods section and in Supplementary Materials. Thus, the resulting part of the paper is not overloaded with technical details. It is not an original organization of the such kind of manuscripts, many analogous papers published in different scientific journals are organized in a same way (please, see refs 19-26 as examples).                 

Point 2. The chemical synthesis methods are generally well-described and seemingly well-executed. They adhere to the norms in the chemical synthesis field with one exception. Typically 13C NMR are reported to 1 decimal place. (i.e. 77.7 ppm).

Response 2. Numbers corresponding to 13C NMR were corrected, see, please, pp. 16 and 18.

Point 3. The biological results are presented clearly for a general audience including in vitro data and histochemical staining. These studies are done for both cytochine and anti-cytokine activity. The results and tables are numerous. Therefore, it may be beneficial to more clearly articulate the hit compounds in the conclusion section using compound numbers. For example, 2 new compounds are alluded to in lines 789-793 but are not referenced by structure or compound number.

Response 3. These data are now referenced by compound names.

Point 4. Clarity:
line 73: They do not overlap because of the different spatial arrangement of the substituents around the optically active (chiral) carbon atom.
*The identity of a structure with its mirror image is somewhat unclear by the statement above. Perhaps: "Their structures are non-identical because..." would be more clear.

Response 4. Accepted.

Point 5. line 77: In nature, ...
*The statements about the inherent handedness of molecules in nature are generally true but are not wholly accurate. The S enantiomer of cysteine is natural while glycine is achiral. Also L-fucose is a common L-sugar in mammals and many other L configured carbohydrates are common in microorganisms. Some minor qualifying terms are necessary to make the claims wholly true.

Response 5. Corresponding corrections in the Introduction were made.

Point 6. Grammar suggestions:
line 19: All compounds contained a methyl group at the α-carbon atom of the benzyl moiety being R- or S-enantiomers.
line 48: Despite their structural similarity, receptors of the same family often have differences in ligand specificity, which seems to be related to their participation in the long-distance root-to-shoot signaling.
Line 102: To modify natural cytokinin N6-benzyladenine (BA) molecule with the α-methylbenzyl
chiral fragment, various 6-chlorosubstituted purines were introduced into the amination reaction with R- and S-(α-methylbenzyl)amine in the presence of Hunig’s base DIPEA
(Scheme 1) as was previously shown for non-chiral purine derivatives [19,20].

Response 6. All grammar suggestions were accepted with gratitude.